# No evidence from complementary data sources of a direct glutamatergic projection from the mouse anterior cingulate area to the hippocampal formation

Lilya Andrianova[1,2,3], Steliana Yanakieva[3], Gabriella Margetts-Smith[1], Shivali Kohli[1], Erica S Brady[1], John P Aggleton[4], Michael T Craig[1,2]*

[1]Institute of Biomedical and Clinical Science, University of Exeter Medical School, Exeter, United Kingdom; [2]School of Psychology & Neuroscience, College of Medical, Veterinary and Life Sciences, University of Glasgow, Glasgow, United Kingdom; [3]School of Infection & Immunity, College of Medical, Veterinary and Life Sciences, University of Glasgow, Glasgow, United Kingdom; [4]School of Psychology, Cardiff University, Cardiff, United Kingdom

**Abstract** The connectivity and interplay between the prefrontal cortex and hippocampus underpin various key cognitive processes, with changes in these interactions being implicated in both neurodevelopmental and neurodegenerative conditions. Understanding the precise cellular connections through which this circuit is organised is, therefore, vital for understanding these same processes. Overturning earlier findings, a recent study described a novel excitatory projection from anterior cingulate area to dorsal hippocampus. We sought to validate this unexpected finding using multiple, complementary methods: anterograde and retrograde anatomical tracing, using antero-grade and retrograde adeno-associated viral vectors, monosynaptic rabies tracing, and the Fast Blue classical tracer. Additionally, an extensive data search of the Allen Projection Brain Atlas database was conducted to find the stated projection within any of the deposited anatomical studies as an independent verification of our own results. However, we failed to find any evidence of a direct, monosynaptic glutamatergic projection from mouse anterior cingulate cortex to the hippocampus proper.

*For correspondence:
mick.craig@glasgow.ac.uk

Competing interest: The authors declare that no competing interests exist.

## Editor's evaluation

In this important study, the authors attempted to further examine the existence of a potential direct projection from the anterior cingulate cortex to the hippocampus which has important functional implications but is currently supported by only limited evidence. The authors used appropriate and complementary approaches to provide compelling evidence for the conclusion that they found no evidence in support of the existence of this connection.

## Introduction

Models of the functional interactions between the rodent prefrontal cortex and the hippocampal formation have had to address their asymmetrical relationship. The subiculum and CA1 give rise to direct efferents that terminate in the prelimbic, infralimbic, and medial orbital cortices (*Cenquizca*

and Swanson, 2007; Groenewegen et al., 1987; Jay and Witter, 1991; Swanson, 1981), as well as light projections to the anterior cingulate area (ACA; Cenquizca and Swanson, 2007; Jay and Witter, 1991). In contrast, direct projections from the prefrontal cortex (including prelimbic and anterior cingulate cortices) have been extensively studied and prefrontal efferents to the Cornu Ammonis (CA) fields or the subiculum using classical tracers are not observed in rat (Heidbreder and Groenewegen, 2003; Jones and Witter, 2007; Segal and Landis, 1974; Sesack et al., 1989; Vertes, 2004; Vogt and Miller, 1983). Meanwhile, only a very sparse projection from infralimbic cortex to CA1 has been described (Hurley et al., 1991), Indeed, a comprehensive connectome was generated using an extensive, curated dataset from published anatomical literature in the rat and reported no evidence of an efferent projection from ACA or prelimbic cortex (PL) to the hippocampal CA subfields (Bota et al., 2015). Similarly, no projection from any prefrontal region to hippocampus (HPC) is observed in marmosets (Roberts et al., 2007). Relatively fewer studies using classical tracers to study efferent connections from mouse prefrontal cortex (PFC) have been carried out. However, it seems that mouse ACA projections correspond well with those seen in rat, with no prefrontal efferents being located in hippocampus proper (Fillinger et al., 2018). For this reason, models of prefrontal regulatory action upon the hippocampus have emphasised the importance of indirect routes. These routes include relays via parahippocampal cortices and via subcortical sites, such as nucleus reuniens and the anterior thalamic nuclei (Eichenbaum, 2017; Furtak et al., 2007; Jones and Witter, 2007; Prasad and Chudasama, 2013).

This relationship was transformed by the description of light, direct projections from the dorsal ACA to the CA1 and CA3 fields in mice (Rajasethupathy et al., 2015) using anterograde adeno-associated viral vector (AAV) and retrograde monosynaptic rabies approaches. Despite their sparsity, optogenetic manipulation of these same projections was sufficient to elicit contextual memory retrieval, and optogenetic stimulation of these axons evoked robust excitatory postsynaptic currents in pyramidal cells in both CA3 and CA1 (Rajasethupathy et al., 2015). These findings are striking, not least because by revealing a previously unknown connection in the rodent brain, they suggest that there are potentially many such functional connections still waiting to be uncovered. A second study, using retrograde AAVs, reported an incredibly sparse population of ACA neurons (1–3 neurons per section) that projected monosynaptically to both dorsal and ventral CA1 (Bian et al., 2019). However, other mouse studies using anterograde AAVs to map efferents from ACA report no projection to hippocampus (Glat et al., 2022; Shi et al., 2022). While not reported explicitly, representative images from another study mapping outputs of ACA revealed no labelling in hippocampus (Zhang et al., 2016). An inhibitory projection from prelimbic and infralimbic regions to hippocampus was recently described (Malik et al., 2022); this projection was reported to selectively target inhibitory interneurons in hippocampus whilst avoiding pyramidal cells, which contrasts with excitatory ACA projection to hippocampal pyramidal cells that was reported in the first study (Rajasethupathy et al., 2015).

Given the impact of their functional findings on anterior cingulate efferents (Rajasethupathy et al., 2015), the present study re-examined the status of the projections from the ACA to the hippocampal formation (dentate gyrus [DG], CA fields, and subiculum [SUB]) in mice. Anterograde tracing using AAVs found no evidence of the anterior cingulate to hippocampus projection in mice, nor did rabies virus-assisted monosynaptic retrograde tracing from hippocampal pyramidal cells. Similarly, classical retrograde tracing using Fast Blue failed to reveal evidence of a direct projection from ACA to hippocampus. Furthermore, a thorough data mining analysis of Allen Expression data also revealed no evidence of an anterior cingulate to HPC projection in mice.

## Results

### Anterograde virus-assisted tracing

We carried out stereotaxic injections of AAV vectors to allow expression of GFP or mCherry in ACA neurons to allow us to map their projections (Figure 1). We targeted our injections into ACA/PFC at two different AP levels: 1.0 mm rostral to Bregma (Figure 1A) and 1.75 mm rostral to Bregma (Figure 1B). The more caudal coordinates exactly matched those reported by Rajasethupathy et al., 2015, which also produced some somatic labelling in M2. We then targeted the site deeper and slightly more lateral within a more rostral point in the rostrocaudal axis, which led to somatic labelling restricted entirely within the subdivisions of the prefrontal cortex (the green depictions of the

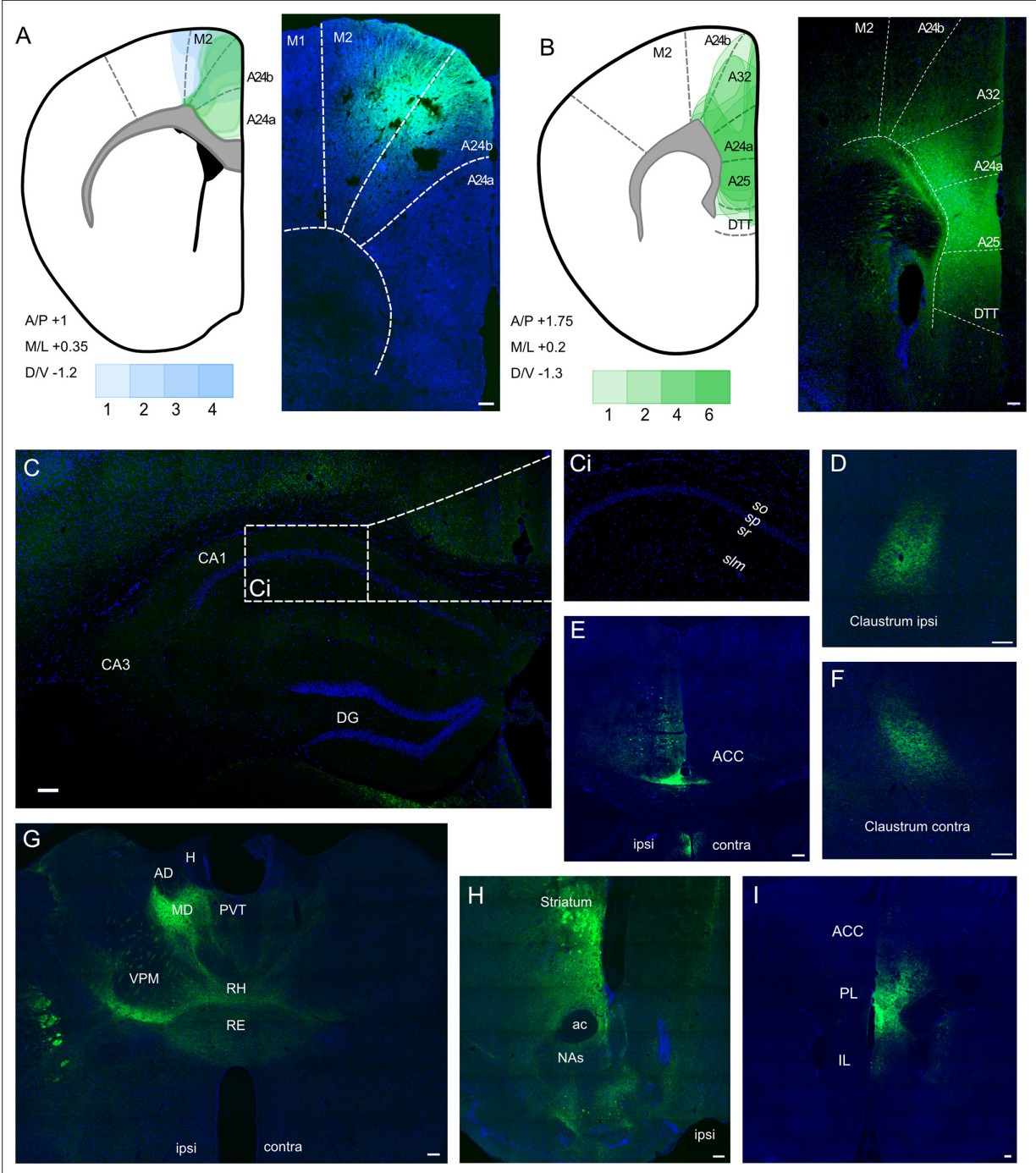

**Figure 1.** Anterograde viral tracing from anterior cingulate area (ACA) CaMKII-expressing neurons shows no excitatory projection to the dCA1 region of hippocampus. (**A**) Prefrontal cortex injection sites and spread, blue overlays show injections made using previously published coordinates (*Rajasethupathy et al., 2015*), green overlays show representative viral spreads of each experimental mouse using in-house initial optimisation attempt, n = 3 and n = 3, respectively. Intensity of colours shows number of animals with the corresponding viral spread, (**B**) Prefrontal cortex injection site and spread of individual injections using in-house optimised coordinates; each overlay shape represents one injection, n = 7. (**C**) Representative images of dCA1 area of hippocampus, Ci closeup image of CA1, showing no fluorescent signal, (**D**) ipsilateral claustrum, (**E**) ACA at +0.8 Bregma, (**F**) contralateral claustrum, (**G**) fibres in some thalamic nuclei, (**H**) fibres striatum, nucleus accumbens lateral septum, and nucleus of the diagonal band, (**I**) fibres in prefrontal cortex at +1.2 Bregma. A24a, area 24a (infralimbic); A24b, area 24b (cingulate cortex 1); A25, area 25 (dorsal peduncular cortex); A32, area 32 (prelimbic); ac, anterior commissure; ACA, anterior cingulate area; AD, anterodorsal nucleus; AM, anteromedial nucleus; AV, anteroventral nucleus; CA1, Cornu Ammonis 1;; CA3, Cornu Ammonis 3; Cl, claustrum; DG, dentate gyrus; H, habenula; LD, lateral dorsal nucleus; M1, primary motor area; M2, secondary motor area; MD, mediodorsal nucleus; NAs, nucleus accumbens shell area; PVT, paraventricular nucleus of the thalamus; RE,

*Figure 1 continued on next page*

*Figure 1 continued*

nucleus reuniens; RH, rhomboid nucleus; VPM, ventral posteromedial nucleus. Transduced neurons and their axons are shown in green while DAPI is shown in blue. Scale bars 100 microns.

The online version of this article includes the following source data and figure supplement(s) for figure 1:

**Source data 1.** Summary of all injections carried out for anterograde tracing from ACA.

**Figure supplement 1.** Comparing viral spread in different injection coordinates of anterior cingulate area (ACA).

viral spread in *Figure 1A*). We further optimised the coordinates as described in the 'Materials and methods' section, and the resulting viral spread can be seen in *Figure 1B*, showing more subregions of the prefrontal cortex, including infralimbic (IL), prelimic (PL), and anterior cingulate (ACC) cortices, have been successfully targeted with minimal spill into the neighbouring regions. Further images of more posterior prefrontal cortices also show clear and strong signal, which is restricted to the injection site: +1.2 (*Figure 1I*) and +0.8 (*Figure 1E*) Bregma, respectively.

A representative image of the dorsal hippocampus can be seen in *Figure 1C*, with *Figure 1Ci* showing a closeup of the CA1, completely devoid of fluorescent signal. The hippocampus was checked across anterior–posterior axis, and it was confirmed that no fluorescent signal was observed at any point. In all cases, we saw prefrontal efferents within several brain regions: namely, both ipsilateral and contralateral claustrum (*Figure 1D and F*, respectively), retrosplenial cortex (*Figure 1C*), several thalamic nuclei (*Figure 1G*), as well as striatum, shell area of nucleus accumbens, lateral septum, and nucleus of the diagonal band (*Figure 1H*), as expected, but found no evidence of axons in any hippocampal subfield or subiculum (*Figure 1C*). *Figure 1—figure supplement 1* presents more examples from the anterograde-tracing experiments, and a summary table of the projections from our anterograde labelling experiment is presented in *Figure 1—source data 1*, which incorporates data from a total of 30 mice.

## Monosynaptic retrograde tracing

After examining with anterograde virus-assisted tracing, we next used monosynaptic rabies-assisted viral tracing to determine whether pyramidal cells in hippocampal region CA1 received direct inputs from ACA (experimental protocol summarised in *Figure 2A*). We carried out tracing from both dorsal CA1 (dCA1; n = 6 mice; *Figure 2B*) and ventral CA1 (vCA1; n = 4 mice; *Figure 2I*). The cells labelled with green fluorescent protein are those that have taken up the 'helper' virus, AAV8-FLEX-H2B-GFP-2A-oG, that was injected into the areas during the first stereotaxic surgery. The cells, expressing the red fluorescent protein, are those that were either transduced with the pseudo-typed rabies virus, EnvA G-deleted Rabies-mCherry, or received the virus via retrograde transsynaptic spread. In this experiment, up the 'starter' cells are those that express both fluorophores and hence appear yellow on the images, as can be seen in *Figure 2B and I* insets.

Acting as positive controls, and as we reported previously (*Andrianova et al., 2021*), retrogradely labelled neurons were detected in all brain regions in all mice that one would expect to project to CA1: both dCA1 and vCA1 received monosynaptic inputs from EC (*Figure 2G and L*). As reported by others (*Sun et al., 2014*), we found evidence of monosynaptic inputs to dCA1 from dorsal subiculum (*Figure 2F*) and to vCA1 from ventral subiculum, although we did not determine whether these neurons were GABAergic or glutamatergic. As expected, vCA1, but not dCA1, received monosynaptic input from the amygdala (*Figure 2N*). Importantly, however, we found no evidence of monosynaptic projections from ACA or any other prefrontal region to either dCA1 (*Figure 2C, D and F*) or vCA1 (*Figure 2J, K, and M*). These data are summarised in *Figure 2P*, and quantification of the number of starter cells can be found in supplementary material (*Figure 2—figure supplement 1*).

## Allen Atlas data

A total of 31 cases had viral injections confined within the ACA. For these cases, we recorded the optical density measures for transported fluorescent fibres in the regions of interest. The medians (*Mdn*) and ranges for each hippocampal region by group (WT or transgenic) are presented in *Figure 3*. In the WT group (*Figure 3A and D*), there was no evidence of signal in either the ipsilateral or contralateral CA1 field (all optical density measures = zero), and only one case contained any signal in the ipsilateral CA3 field (0.0002). For the transgenic group (*Figure 3B–D*), the median signal for CA3

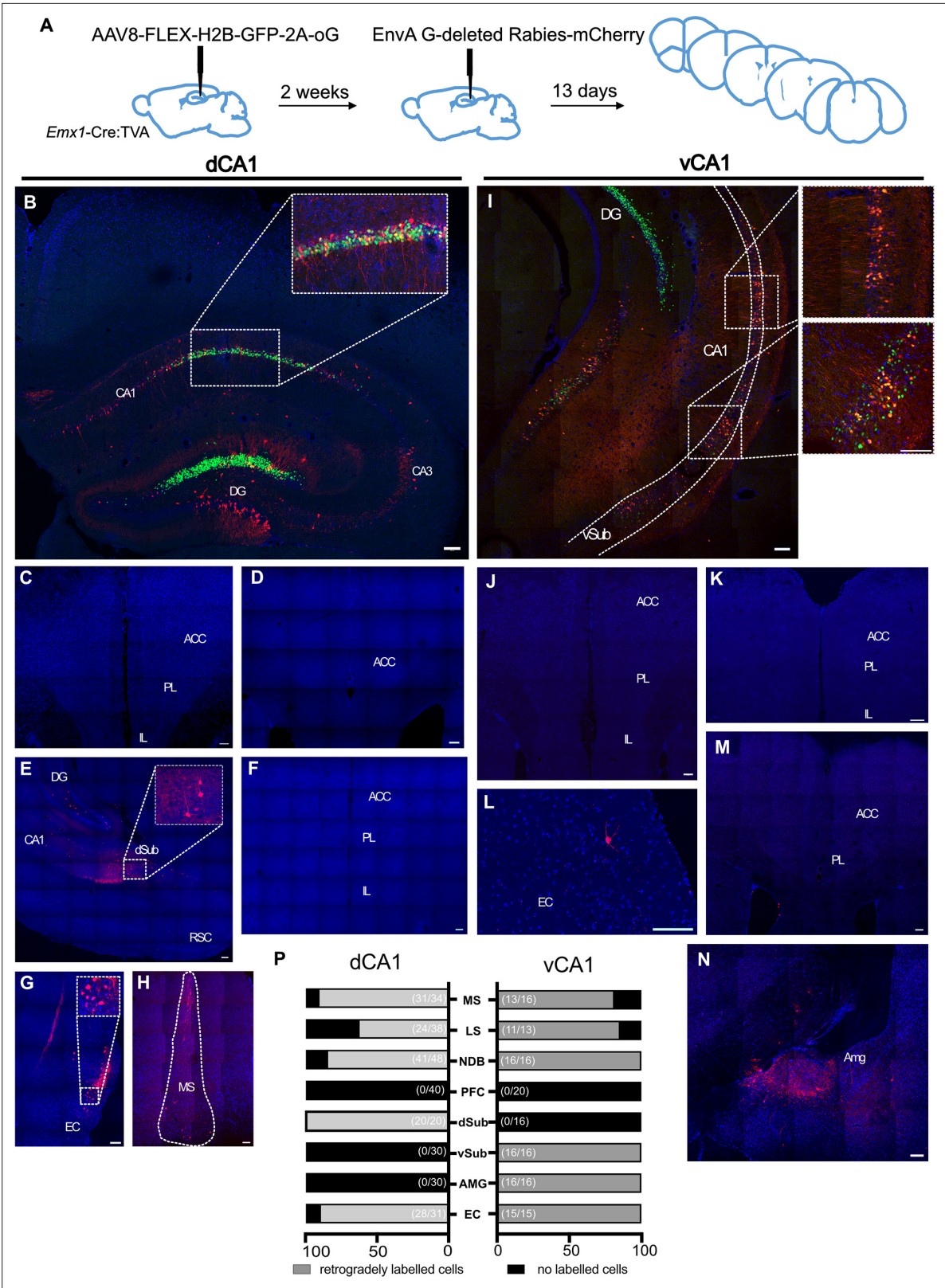

**Figure 2.** Monosynaptic retrograde tracing from dorsal CA1 (dCA1) and ventral CA1 (vCA1) using modified rabies virus shows no excitatory projection from the anterior cingulate area (ACA) to either dorsal or ventral CA1 regions. (**A**) Outline of experimental method. (**B**) Representative images of dCA1 injection site, starter cells shown as inset. Representative images of (**C–F**) prefrontal cortical areas at +1.6, +1.1, and +2.0 Bregma, (**G**) entorhinal cortex, (**H**) medial septum. (**I**) vCA1 injection site, starter cells shown as insets. Representative images of: (**J, K, M**) prefrontal cortical areas at +1.6, +2.0, and

*Figure 2 continued on next page*

*Figure 2 continued*

+1.1 Bregma, (**L**) entorhinal cortex, (**N**) amygdala, and (**P**) summary of data. Bar graphs show the mean percentage of sections from each animal showing presynaptic neurons while numbers in bracket show total number of sections examined. Scale bars represent 100 microns.

The online version of this article includes the following figure supplement(s) for figure 2:

**Figure supplement 1.** Rabies virus tracing – injection sites and starter cells.

and CA1 remained at zero, but there were eight cases with potential signal in the ipsilateral and/or the contralateral CA1 and/or CA3 fields (all <0.0007). However, in all of these cases there was a low, but variable, background signal. For this reason, we compared the signal in areas CA1 and CA3 (which have potential anterior cingulate inputs) with that in the DG and area CA2 (neither of which are thought to receive anterior cingulate inputs). To maximise the likelihood of finding evidence for a projection, we just used the eight cases with possible CA3 or possible CA1 label for the statistical comparisons.

There was no evidence that either CA1 or CA3 contained higher signal levels in these eight cases, that is, the higher optical measures reflected a higher background signal. Wilcoxon signed-rank tests found no statistical difference in the signal between the ipsilateral DG and CA3 ($z = -0.845$, p=0.388), CA2 and CA3 ($z = -0.510$, p=0.610), or the contralateral DG and CA3 ($z = -0.755$, p=0.438), and contralateral CA2 and CA3 ($z = -1.841$, p=0.066). Likewise, there were no significant differences between the signal in ipsilateral CA2 and CA1 ($z = -1.63$, p=0.102) or the contralateral CA2 and CA1 ($z = -1.07$, p=0.285). While there were significant differences between both the ipsilateral DG and CA1 ($z = -2.4$, p=0.016) and the contralateral DG and CA1 ($z = -2.2$, p=0.028), inspection of the data showed that in seven of the cases the signal was higher in the DG than CA1, that is, the opposite direction to that consistent with an ACA projection to CA1.

In contrast, the signal in the cingulum bundle (both ipsilateral and contralateral) was higher in 29 of the 31 cases than the signal in CA1-CA3 and the DG. In the WT group, the median signal density in the ipsilateral CB was 0.031 and 0.0014 in the contralateral CB 0.014 (see *Figure 3Ei* for ranges). Similarly, the transgenic group with right-hemisphere injections had a median signal density in the ipsilateral CB of 0.0043 and 0.0002 in the contralateral CB (*Figure 3Ei*). The presence of CB fibre label helps to confirm the effective transport of the tracer from the injection site. Lastly, there was no evidence of a projection from the ACA to the subiculum in any group as the median signal density observed for the ipsilateral and contralateral subiculum was a maximum of 0.0001 (*Figure 3Eii*).

The median density signal in the 16 transgenic cases with injection volumes of over 90% within the ACA was consistent with the data from the other 28 transgenic cases (see *Figure 1—source data 1*). The median group signal in both CA1 and CA3 was zero, with 10 cases having potential evidence of a signal (max = 0.0007) in CA1 and/or CA3. Again, Wilcoxon signed-rank tests revealed no evidence of a statistical difference in the signal between the ipsilateral or contralateral DG and CA3 or CA2 and CA3 (*all $z < -1.63$, all $p_s > 0.102$*) nor between the ipsilateral and contralateral CA2 and CA1 (*all $z < -1.34$, all $p_s > 0.180$*). There were, however, statistically significant differences between both the ipsilateral DG and CA1 ($z = -2.56$, p=0.010) and the contralateral DG and CA1 ($z = -2.03$, p=0.042), but again the higher signal was in the DG. Further inspection of individual cases showed that one case (ID: 125801033) had noticeably higher signal density in the ipsilateral DG (0.0108) and SUB (0.0039). A second case (ID: 585911240) had the highest signal density in the ipsilateral CA3 (0.0007). However, in both cases 5% of the injection had leaked into adjacent fibre tracts.

Of concern was whether the adjacent cortical field M2 might project to the hippocampal formation and, thereby, be a potential confound for tracer injections that extended into this area. For this reason, we examined the *Allen Mouse Brain Connectivity Atlas* (http://mouse.brain-map.org/) for wild-type cases meeting the inclusion criteria with injections of 90% or more within area M2. Any case in which the tracer extended into the ACAs was excluded. The search returned only one such case (ID: 585025284), which showed no evidence of a projection to CA1, CA2, or subiculum (all signal density counts zero). There was a very low signal count within the ipsilateral CA3 and DG (<0.0002). In an additional case (ID: 141602484), the viral injection involved both the primary motor cortex (46%) and area M2 (54%). The only hippocampal signal was in the ipsilateral CA3 (<0.0002). Both cases had appreciably more signal in the ipsilateral than contralateral cingulum bundle, consistent with viable transport.

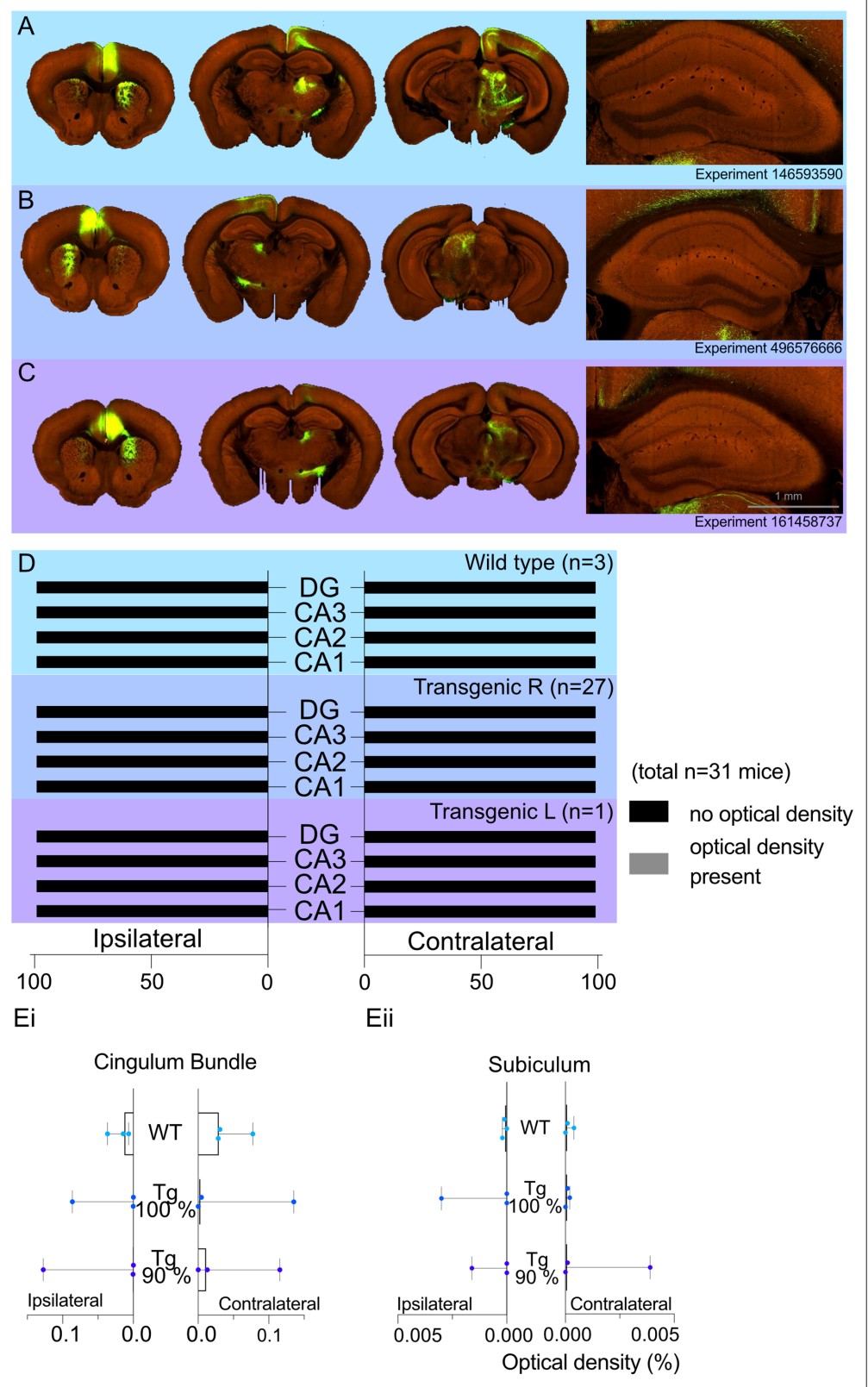

**Figure 3.** Allen Projection Atlas data search shows no evidence of a direct projection from anterior cingulate area (ACA) to dorsal CA1. Representative examples of tracing experiments in (**A**) wild-type mouse line (C57BL/6J), (**B**) transgenic (Rbp4-Cre_KL100) mouse line experiment with 100% of the virus injected in the ACA, (**C**) transgenic (Rbp4-Cre_KL100) mouse line experiment with 90% of the virus injected in the ACA; and the corresponding typical

*Figure 3 continued on next page*

*Figure 3 continued*

anterograde expression patterns. (**D**) Histograms showing the summary of fluorescent fibres present in different hippocampal subfields in all datasets. No fibres were found any sections containing hippocampal regions DG and CA1-3. (**E**) Median signal densities of fibres seen in cingulum bundle (**Ei**) and subiculum (**Eii**) in experiments with wild-type and transgenic mouse lines; box plots show median and error bars the range of the data. The histograms show the optical densities of potential transported tracer in different experiments in the cingulum bundle and subiculum. While there is no apparent signal in the subiculum, the tracer uptake is significantly higher in the cingulum bundle.

The online version of this article includes the following source data and figure supplement(s) for figure 3:

**Source data 1.** Numerical data from Allen Projection Atlas mining that was used to generate figure 3.

**Figure supplement 1.** Images from prefrontal projectome paper (*Gao et al., 2022*).

## Retrograde tracing using a single virus strategy

Thus far, we have failed to detect evidence of a direct projection from the cingulate area to hippocampus. The question remains of why there is a discrepancy between our data and that reported previously (*Rajasethupathy et al., 2015*). In our rabies-virus assisted retrograde tracing experiment, we used a pseudotyped glycoprotein-deleted rabies virus that could only transduce neurons expressing the avian receptor protein, TVA, and could only spread transsynaptically from glutamatergic neurons that also expressed the rabies glycoprotein, delivered via a secondary helper virus, as described elsewhere (*Sun et al., 2014*). In contrast, *Rajasethupathy et al., 2015* used a non-pseudotyped glycoprotein-deleted rabies virus which acts as a first-order tracer from the injection site (*Wickersham et al., 2007*). To test whether this strategy could provide off-target retrograde labelling, we made stereotaxic injections of a retrograde AAV vector (*Tervo et al., 2016*) into dorsal CA1 (*Figure 4A*). Using this approach, in addition to dense neuronal labelling of CA1 pyramidal cells (*Figure 4B*), we saw labelling in the entorhinal cortex (*Figure 4I*), as would be expected. However, similar to the extended Figure 1 of *Rajasethupathy et al., 2015*, we saw labelling of neurons directly above the injection site, and throughout both superficial and deep cortical layers in much of the brain (*Figure 4B*; see also Extended Data Figure 1a of *Rajasethupathy et al., 2015*). Additionally, we observed dense labelling of neurons in the anterior thalamic nuclei (*Figure 4D*), which do not project to hippocampus proper (*Mathiasen et al., 2020*) and, importantly, we saw bilateral labelling in multiple subdivisions of prefrontal cortex, including the ACA (*Figure 4E, H and G*); it was most dense in the ipsilateral hemisphere (*Figure 4E*). We also observed fluorescent labelling in the claustrum (*Figure 4F*), which should only project to cortical areas. Without the selectivity of the pseudotyped rabies-tracing approach from specific neuronal populations, we believe that the off-target retrograde labelling in this region was due to uptake of virus in axons passing through the cingulum and/or corpus callosum as well as along the path of the injection needle.

## Retrograde tracing using a Fast Blue dye

We also used a classical retrogradely transported dye, Fast Blue, to confirm the presence or absence of this direct projection. In the total of 11 injections with 3% dilution of the dye, 4 brains displayed substantial uptake by neurons in the hippocampus while only limited spillage of the dye into the overlaying cortex was observed. These brains were added to the experimental total. One mouse was excluded as few fluorescent cells were present in the injection site, most likely due to partial blockage of the Hamilton needle during the injection. Several mice showed the spread of the injection site beyond the hippocampus into surrounding areas, including the dorsal subiculum and somatosensory cortex; in these animals, we saw a high prevalence of fluorescent cells across throughout the cortex, in different subregions of the prefrontal cortices, from infralimbic to anterior cingulate cortices, as well as thalamic nuclei. The example of such spread can be found in supplementary material (*Figure 5— figure supplement 1*). In the four good injections, where the dye was restricted primarily to the dorsal CA1 region of the hippocampus (*Figure 5B*) and only some cells were labelled along the travel path of the needle, there were no labelled cells present in the prefrontal cortices, including ACC, PL, and IL subregions regions (*Figure 5C and D*). However, other regions, known to project to dorsal CA1, such as medial septum, dorsal subiculum, and entorhinal cortex all showed dye-filled cells (*Figure 5F, G*

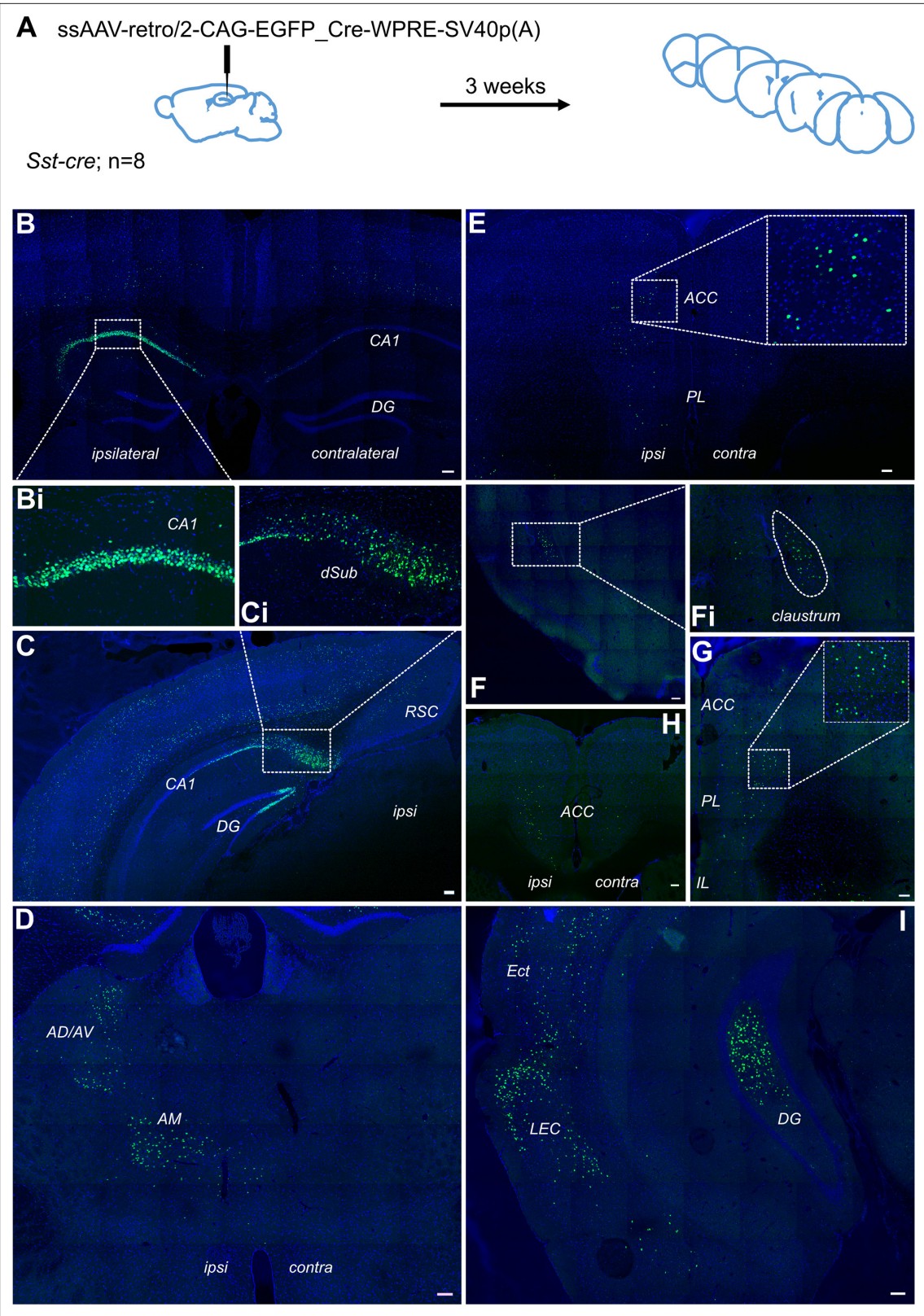

**Figure 4.** Retrograde tracing from dCA1 shows non-specific expression in multiple brain area, including both ipsi- and contralateral anterior cingulate and prelimbic areas. (**A**) Outline of experimental method. Representative images of (**B**) injection site dCA1, showing strong labelling in the hippocampus as well as some labelling in the overlaying cortical areas, including contralateral cortex, (**Bi**) cells in CA1 pyramidal layer, (**C**) dorsal subiculum (dSub), (**Ci**) neurons in dSub, (**D**) retrograde labelling in anterodorsal and anteroventral (AD/AV) and anteromedial (AM) thalamus nuclei, (**E, H, G**) labelled cells

*Figure 4 continued on next page*

*Figure 4 continued*

present in both hemispheres of prefrontal cortices at +1.7, +1.2, and+ 1.5 Bregma showing prelimbic (PL) and anterior cingulate cortex (ACC), (**F**) cells in claustrum, (**I**) cells in lateral entorhinal (LEC), perirhinal and ectorhinal (Ect) cortices. Scale bars 100 microns.

*and E*, respectively). Thus, we conclude that using this classical tracer method we are not able to find any direct projection from the ACC to dorsal CA1.

## Discussion

Here, we attempted to replicate the report of a novel projection from ACA to the hippocampus (*Rajasethupathy et al., 2015*) but failed to find any evidence of this projection in the mouse after examining a combination of tracing methods. Importantly, both our retrograde and anterograde-tracing experiments revealed the existence of expected efferent projections from ACA and afferent inputs to hippocampus. For example, our anterograde AAV tracing from ACA revealed projections to midline thalamic regions, retrosplenial cortex, claustrum, and striatum, in agreement with classical tracing experiments from rat (summarised by *Bota et al., 2015*), mouse (*Fillinger et al., 2018*), and primate (*Roberts et al., 2007*). Similarly, our monosynaptic rabies-tracing experiments found inputs to hippocampus from regions such as medial septum and entorhinal cortex, in agreement with a broad wealth of literature (reviewed by, e.g. *Witter et al., 1989*). We have presented wide-field images from all of our tracing experiments to confirm the specificity of the projections being described. In cases where experiments do not work as planned due to a lack of specificity of the tracer, the presentation of wide-field images makes this immediately apparent and urges caution when interpreting the results. We would recommend that, when describing a novel neural projection, authors also include control data to reassure readers that known projections are also present but without widespread non-specific labelling.

### Retrograde tracing from hippocampus

We employed an intersectional approach with rabies viruses using an EnvA-pseudotyped rabies virus combined with TVA expression restricted to hippocampal pyramidal cells via the *Emx1* promoter. As a result, this reduced our density of starter cells, so the possibility remains that some sparse projections would be missed, or those that specifically target inhibitory interneurons such as the inhibitory prefrontal projection reported by *Malik et al., 2022*. However, we complemented our dataset with the classical tracer, Fast Blue, the results of which supported our viral tracing findings. It should be noted that the Fast Blue experiments did not retrogradely label any neurons in PL or IL. The postsynaptic targets of the reported inhibitory projection neurons in PFC primarily target deeper layers within CA1 and, as can be seen from *Figure 5A*, our Fast Blue injections primarily hit *stratum oriens* and *stratum pyramidale*, potentially explaining why we did not observe an inhibitory projection.

Despite these caveats associated with our present study, we believe that this approach substantially reduces the risk of observing false positives. In contrast, the study using an un-pseudotyped rabies virus (*Rajasethupathy et al., 2015*) displayed examples of what appears to be off-target expression in a number of brain regions, such as in the secondary motor cortex, retrosplenial cortex, and possibly somatosensory cortex. Notably, specific transduction of corpus callosum oligodendrocyte precursor cells with rabies virus reveals that these cells receive widespread presynaptic input from PFC, motor cortex, and somatosensory cortex (*Mount et al., 2019*). Indeed, the distribution of presynaptic somata shown in figures 1 and 2 of the Mount study bears a striking resemblance to that shown in *Rajasethupathy et al., 2015*. By using a rabies virus able to transduce any neural cell, the possibility exists that at least some of the retrogradely labelled neurons were due to off-target transduction of oligodendrocyte precursor cells overlying the hippocampal injection site.

The retrograde tracing from the Bian (*Bian et al., 2019*) produced far more sparse labelling than that reported by *Rajasethupathy et al., 2015*. This later study used a retrograde AAV strategy that, from the figures shown, produced a sparse labelling of neurons in ACA and secondary motor cortex, with only two or three somata present per brain region. The titre of retroAAV used in the study was not stated but the injection volume was 400 nl in ACA and 600 nl in hippocampus; the low density of neurons transduced in the injection sites (e.g. figure 5A) is surprising (*Bian et al., 2019*). This study also failed to detect the vHPC to ACA projection that has been described in classical tracer studies

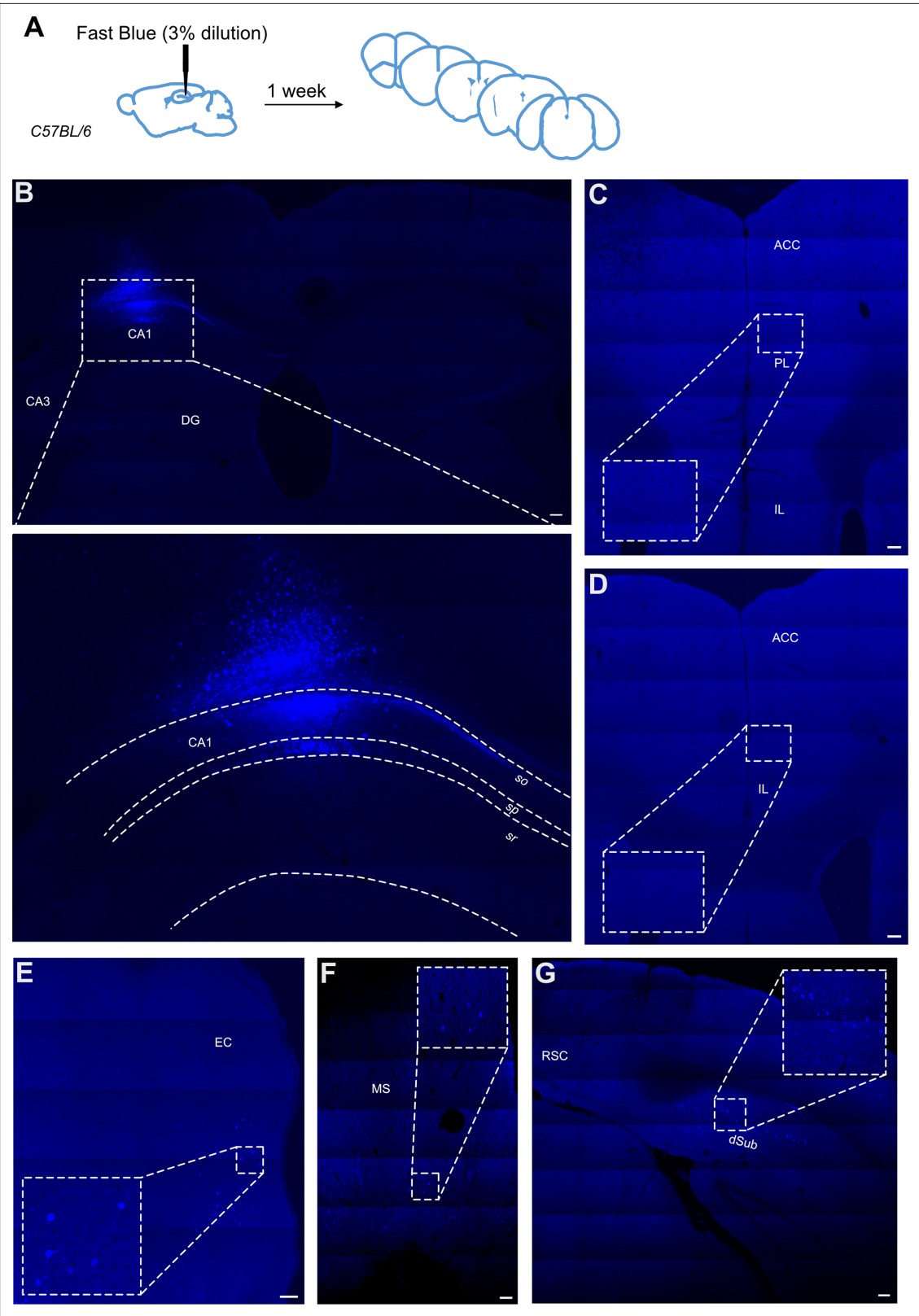

**Figure 5.** Retrograde tracing from dCA1 using Fast Blue reveals no positive labelled neurons in the anterior cingulate area (ACA). (**A**) Outline of experimental, (**B**) injection site and inset showing filled cells in the CA1 region of hippocampus and some overlaying cortex, (**C, D**) representative images showing no filled cells in prefrontal cortices at +1.4 and +1.1 Bregma, insets show close up of PL and ACC areas, (**E**) filled cells in entorhinal cortex (EC), (**F**) filled cells in medial septum (MS), and (**G**) filled cells in dorsal subiculum (dSub). RSC, retrosplenial cortex, N = 7. Scale bars 100 microns.

*Figure 5 continued on next page*

*Figure 5 continued*

The online version of this article includes the following figure supplement(s) for figure 5:

**Figure supplement 1.** Retrograde tracing from dCA1 shows positive cells in the anterior cingulate area (ACA) when injected dye spreads off target.

(*Cenquizca and Swanson, 2007*; *Hoover and Vertes, 2007*; *Jay and Witter, 1991*). It should be noted that at least one study of ventral hippocampal projections to PFC found preferential targeting of prelimbic and infralimbic regions over ACA (*Spellman et al., 2015*), although this later study was carried out in rostral PFC, and the hippocampal axon termination patterns shown correspond to those with classical tracer studies (*Cenquizca and Swanson, 2007*; *Hoover and Vertes, 2007*; *Jay and Witter, 1991*). The region of ACA targeted by Bian and colleagues was from a caudal region of ACA where the classical tracer studies show the presence of efferents from ventral hippocampus.

## Anterograde tracing from ACA

We also attempted to find evidence of a direct glutamatergic projection from ACA to HPC using anterograde tracing with AAV vectors of excitatory opsins conjugated to a fluorophore. In contrast to the data reported by *Rajasethupathy et al., 2015*, we did not observe any ACA axons in HPC, which is in agreement with other subsequent anterograde-tracing studies using the same approach (*Fillinger et al., 2018*; *Wang and Ikemoto, 2016*). A more recent study employed the fMOST technique to map the whole-brain projectome of individual neurons from different prefrontal areas, mapping the complete axonal arborisation of 6357 glutamatergic neurons in PFC, and clustering these into 64 specific subtypes (*Gao et al., 2022*). The dataset from this study is available online (https://mouse.braindatacenter.cn/), and we searched the dataset for PFC neurons with at least two axon termination points in CA hippocampal subfields; criteria recommended by the authors. Of the 6357 neurons, we found only four PFC neurons that had terminations in the hippocampus, and only three of which had terminations in dHPC (see *Figure 3—figure supplement 1*). It should be noted that none of these prefrontal HPC-projecting neurons were located in ACA, with somata located in lateral orbitofrontal cortex (layer 2/3; two neurons), anterior secondary motor cortex (layer 5; one neuron), and ventral agranular insular cortex (one neuron; layer 2/3) with most terminations near the hippocampus located just dorsal of the CA subfields. Despite tracing the complete projectome of over 6000 PFC neurons, the study by *Gao et al., 2022* did not provide any evidence to support the existence of a direct ACA to HPC glutamatergic projection.

We are unable to provide an explanation for the discrepancy between the findings of *Rajasethupathy et al., 2015* and those from our and other research groups. The injection spread of their anterograde-tracing experiment, shown in *Figure 1b* (*Rajasethupathy et al., 2015*) clearly includes the cingulate area, but also the M2 part of motor cortex is transduced by the virus; it is also possible that the transitional midcingulate area 24' (*Vogt and Paxinos, 2014*) has been targeted. Indeed, when using the reported coordinates, we again found significant spread of the virus in M2. Furthermore, looking at the projection pattern of the anterograde fibres in *Rajasethupathy et al., 2015*, it can be noticed that both hippocampi show fluorescent fibres, with the fibres in the contralateral hippocampus (right; *Figure 1b* in their study) being brighter than in the ipsilateral hippocampus; the pattern, opposite to that in the case of the thalamus. Having used 250 nl of virus, compared with 500 nl from *Rajasethupathy et al., 2015* we still observe the spillover into neighbouring brain areas with these coordinates. Whilst the spillover of the virus itself cannot account for the apparent presence of axons from the cingulate area to CA1, nor the optogenetic activation of these afferents in that study, one potential explanation of a false positive could be that the relatively large volume of virus with a high titre injected close to the midline led to the leakage of virus beyond the intended injection site, resulting virus being taken up via axons traversing the cingulum bundle.

Should there exist a direct projection from ACA to HPC that our group and others have failed to detect, one possible explanation could be a difference in the tropism of viral vectors used to transduce neurons, with the ACA to HPC-projecting neurons uniquely resistant to transduction by vectors that are able to target other neurons within the same brain region. Another explanation could be the incubation time for viral expression from the time of injection to perfusing the animals, with our study using (at least) 3 wk, compared to the 30-day (anatomical tracing) or 6–8-week (optogenetics) incubation used by *Rajasethupathy et al., 2015*. However, strong AAV-driven gene expression in vivo is typically seen after 2–3 wk (e.g. *Aschauer et al., 2013*; *Stoica et al., 2013*; *Xu et al., 2001*), and the

fMOST PFC projectome study used 4-week incubation (*Gao et al., 2022*). Intuitively, it seems unlikely that the fMOST study would reveal PFC axonal projections that are distributed across the entire brain after 4 wk, while selectively sparing the labelling of axons that enter HPC. And these viral tropism/incubation hypotheses cannot account for why any ACA to PFC projection had remained undetected using classical tracer methods.

### Concluding remarks

Here we report that we have been unable to find evidence of a monosynaptic glutamatergic projection from ACA to hippocampus proper in the mouse, replicating findings reported over decades of rodent neuroanatomical literature but in stark contrast to two recent reports (*Bian et al., 2019*; *Rajasethupathy et al., 2015*). While we have presented some tentative explanations that could account for these discrepancies, we remain unable to provide a full account of why we and others failed to find evidence of this projection, should it exist. Our intention in reporting our findings is not to criticise the research carried out by colleagues, but to initiate a wider conversation about how we use genetic tools in systems neuroscience and interpret the data. The hippocampal–prefrontal system is perhaps one of the most widely studied long-range circuits in cognitive neuroscience; that controversies still exist on the precise cellular connections between these two areas serves to highlight both the excitement and challenges that arise as modern genetic tools provide us with opportunities to study neural circuitry in detail that was unimaginable just 20 years ago.

## Materials and methods

All UK-based research was carried out in accordance with the UK Animals (Scientific Procedures) Act 1986 and was subject to local ethical review by the Animal Welfare and Ethical Review Board at the University of Exeter or University of Glasgow. All animals were maintained on a 12 hr constant light/dark cycle and had access to food and water ad libitum. We used standard enrichment that included cardboard tubes, wooden chew blocks and nesting material.

### Stereotaxic coordinates and the borders of anterior cingulate area

Prefrontal cortex has been extensively studied in multiple model organisms, from mice and rats to humans and nonhuman primates. However, despite a large amount of homology across species, the shape of the prefrontal areas in the brains of different species is quite varied. These variations in shape, together with distinctions in the cytoarchitecture of the subdivisions of the prefrontal cortex, have led to a lack of agreed nomenclature and clear boundaries of specific prefrontal regions (see *Laubach et al., 2018* for a comprehensive analysis of discrepancies between authors). In this article, we use the nomenclature and delineation from *Vogt and Paxinos, 2014* and the human homologies (Brodmann's areas). A very helpful online unified anatomical atlas, combining the Franklin-Paxinos and the common coordinate framework from the Allen Institute of Brain Science, was also used cross-check the anatomical borders of the prefrontal areas (*Chon et al., 2019*).

### AAV anterograde and retrograde tracing

This tracing study used adult C57BL/6 (Envigo, UK) or *Sst-cre* (Jackson Laboratories, USA; stock number 018973) mice of both sexes aged 2–5 mo, ranging from 22.2 g to 31.3 g (mean age 3 mo, mean weight 25 g). To target the ACA, we made stereotaxic injections of 250 nl of AAV5-CamKII-GFP $5.3 \times 10^{12}$ vg/ml (Viral Vector Facility, Neuroscience Centre Zurich, Switzerland) or AAV5/2-hSyn1-ChR2_mCherry $7.1 \times 10^{12}$ vg/ml (Viral Vector Facility, Neuroscience Centre Zurich). Briefly, mice were anaesthetised with 5% isoflurane, were placed on a heated pad for the duration of the surgery, and maintained at 1.5–2.5% isoflurane (with a flow rate of ~2 l min$^{-1}$ O$_2$). Mice were given 0.1 mg/kg of buprenorphine (buprenorphine hydrochloride, Henry Schein) subcutaneously at the start of surgery as an adjunct analgesic, and carprofen (Rimadyl, Henry Schein) was given at a dose of 5 mg/kg subcutaneously at the end of surgery and on subsequent days, as required. An incision was made down the midline, and a craniotomy was performed to allow injection of virus (250 nl). For anterograde tracing, we first used the coordinates reported by *Rajasethupathy et al., 2015*: A/P +1.0 mm (relative to Bregma), M/L –0.35 mm, and D/V-1.2 mm (from pia). Additionally, to minimise spread of virus into M2 cortical area, we also injected at A/P +1 mm (relative to Bregma), M/L –0.2 mm, and D/V-1.3 mm (from pia) on the

same rostrocaudal plane, and targeted ACA at a more rostral site A/P +1.75 mm (relative to Bregma), M/L –0.2 mm, and D/V –1.5 mm (from pia). For retrograde tracing with AAVs, we used rAAV-retro, developed at HHMI Janelia Campus (*Tervo et al., 2016*), and injected rAAV-retro/2-CAG-EGFP_Cre-WPRE-SV40p(A) 5.9 × $10^{12}$ vg/ml (Viral Vector Facility, Neuroscience Centre Zurich) into hippocampal region CA1 using the following co-ordinates: A/P –2 mm (relative to Bregma), M/L –1.5 mm, and D/V –1.35 mm (from pia). After surgical repair of the wound, mice were given 5 mg/kg carprofen (Rimadyl, Henry Schein) subcutaneously immediately after surgery and again the following day. Further analgesia was provided as required. The mice were maintained for at least 3 wk to provide sufficient time for expression of virally delivered transgenes and were killed by transcardial perfusion/fixation with 4% paraformaldehyde (Sigma-Aldrich, UK) in 0.1 M phosphate buffer.

Perfused brains were cryoprotected with 30% sucrose solution (Fisher Scientific, UK) and sliced at 30 µm using a SM2010R freezing microtome (Leica, UK). The slices were stained with DAPI (HelloBio, UK), and images were taken on SLM710 confocal microscope (Zeiss). For anterograde tracing experiments, 32 mice were injected (two were excluded due to failed injections as no fluorescence was observed) and 10 mice were injected with a retrograde virus (two mice were excluded due to failed injections). Viral expression patterns of anterograde viral vectors in individual mice at the injection sites of both sets of coordinates were overlaid (*Figure 1A and B*) to show the variation in the injections and typical spread of the virus in the injection areas. A qualitative assessment of the projection patterns was performed, and representative images of regions of interest are presented in *Figure 1C–I*. Retrograde viral spread in injection site is shown in *Figure 3B*, and images showing the projection pattern can be found in *Figure 4C–I*.

## Monosynaptic retrograde tracing

For monosynaptic rabies tracing, we used methods reported by others (*Sun et al., 2014*). Adult *Emx1-cre* mice (Jackson Laboratories; stock number 005628) (*Guo et al., 2000*) were crossed with floxed TVA mice (kindly provided by Prof Dieter Saur, Technical University of Munich) (*Seidler et al., 2008*) to allow specific targeting of pyramidal cells; *Emx1-cre* mice alone were used as controls to ensure the rabies viruses did not transduce neurons in the absence of TVA gene. We used mice of both sexes aged 3–7 mo, ranging from 21.7 g to 40.6 g (mean age 4.5 mo, mean weight 27.2 g). To target the efferent projections to dCA1, we made stereotaxic injections of AAV8-FLEX-H2B-GFP-2A-oG, titre 3.93 × $10^{12}$ vg/ml (Salk Institute Viral Vector Core) followed by injection EnvA G-deleted Rabies-mCherry 6.13 × $10^{8}$ vg/ml (Salk Institute Viral Vector Core or Charité-Universitätsmedizin Berlin Viral Vector Core) 2 wk after the initial viral injection. Briefly, mice were anaesthetised with 5% isoflurane, were placed on a heated pad for the duration of the surgery, and maintained at 1.5–2.5% isoflurane (with a flow rate of ~2 l $min^{-1}$ $O_2$). Mice were given 0.1 mg/kg of buprenorphine (buprenorphine hydrochloride, Henry Schein) subcutaneously at the start of surgery as an adjunct analgesic, and carprofen (Rimadyl, Henry Schein) was given at a dose of 5 mg/kg subcutaneously at the end of surgery and on subsequent days, as required. An incision was made down the midline and a craniotomy performed to allow injection of virus (250 nl). We targeted hippocampal region CA1 at dorsal and ventral points using the following coordinates: A/P –2 mm (relative to Bregma), M/L –1.5 mm, and D/V –1.35 mm (from pia) and A/P –2.8 mm (relative to Bregma), M/L –2.4 mm, and D/V –4.2 mm (from pia) for dorsal and ventral CA1 regions, respectively. After surgical repair of the wound, mice were given 5 mg/kg carprofen (Rimadyl, Henry Schein) subcutaneously immediately after surgery and again the following day. Further analgesia was provided as required. The mice were maintained for 2 wk to provide optimal time for expression and were killed by transcardial perfusion/fixation with 4% paraformaldehyde (Cat# P6148, Sigma-Aldrich, UK) in 0.1 M phosphate buffer.

Perfused brains were cryoprotected with 30% sucrose solution (Fisher Scientific) and sliced at 50 µm using a SM2010R freezing microtome (Leica). The slices were mounted using HardSet Mounting Medium with DAPI (Vector Labs) and images were taken on SLM710 confocal microscope (Zeiss). A total of 18 mice including four controls were injected (two were excluded due to failed injections as no fluorescence from both viral constructs was observed). Representative images of the spread of the viruses can be found in *Figure 2*. A semi-quantitative assessment of the projection patterns was performed by recording the number of slices with the cells expressing the virus in each region of interest present and the summary of this data is shown (*Figure 2P*). Estimated number of starter cells per mouse was calculated by extrapolating the number of starter cells per section onto estimated viral

spread (*Figure 2—figure supplement 1*). These monosynaptic rabies tracing data were generated in the same experiment that we have reported elsewhere, focusing on connections between nucleus reuniens and CA1 pyramidal cells (*Andrianova et al., 2021*).

### Fast Blue dye retrograde tracing

For classical retrograde tracing, we used Fast Blue dye (3 or 0.6% dilution), as described previously (Yanakieva 2022). Mice of both sexes aged between 4 and 9 months old, ranged from 23.8 g to 36.8 g (mean age 6.2 mo and mean weight 30.4 g). We used coordinates described above to target dCA1. Fast Blue dye (Polysciences, CAS# 73819-41-7) was diluted to 3 or 0.6% working solution using sterile saline and 300 nl were injected into dCA1. Stereotaxic surgery was done as above, and mice were allowed to recover for 7 d before a transcardial perfusion was performed as described above. Cryoprotected brains were sliced at 50-micron thickness on the freezing microtome and visualised using the LSM710 confocal microscope (Zeiss). A total of 15 mice were used and 3 were excluded due to failed injection and a further 5 were excluded due to spill over of dye into overlaying structures. Representative images are shown in *Figure 5* and *Figure 5—figure supplement 1*.

### Allen Projection Atlas data mining

The *Allen Mouse Brain Connectivity Atlas* (http://mouse.brain-map.org/) is a comprehensive database of images of axonal projections in the mouse brain. To create the atlas, each mouse brain is injected with enhanced green fluorescent protein (EGFP) expressing AAV as an anterograde tracer into a *source* brain region (for further details, see http://connectivty.brain-mpa.org). Then, the axonal projections are systematically imaged using a TisseCyte 1000 serial two-photon tomography system. The data in the atlas were collected from adult mice on postnatal day P56 ± 2 (for details, see *Oh et al., 2014*). Each case in the atlas contains high-resolution images and quantified projection information based on the optical density of label. Detailed histograms of the signal in each structure are presented giving the projection volume ($mm^3$) and projection density (the fraction of area occupied by a fluorescent signal from the viral construct used relative to the whole structure). Details of the algorithms used are provided (*Kuan et al., 2015*). The sensitivity of this viral tracer has been compared with biotinylated dextran amine (BDA). These comparisons revealed that the AAV was at least as efficient as and more specific than BDA (http://connectivty.brain-mpa.org).

To establish whether there are direct projections from the ACA (dorsal ACA and ventral ACA) to the hippocampal formation (CA 1–3, DG, and SUB), a comprehensive systematic search was completed. In addition, the density of fibre label in the cingulum bundle (CB) was taken from the Allen Atlas. The dorsal and ventral ACA were entered as a *source structure* and the CA1-3, DG, SUB, and cingulum bundle as target structures. The search was further filtered for (1) reporter type: EGFP; (2) hemisphere: either; (3) minimum target injection volume: $0.0001\ mm^3$. A total of 99 cases were returned. Then, 63 cases with injection volumes of $0.02–0.3\ mm^3$ in the ACA were selected for further processing, of which 47 had a minimum of 90% of the injected virus (mean = $0.108\ mm^3$; SD = $0.059\ mm^3$) within the dorsal and ventral ACAs. *Figure 3—source data 1* lists all 47 cases and provides further information about the transgenic lines and the individual injections. The large majority (n = 44) of the injections were made in the right hemisphere, the remaining three in the left. In the few cases where some signal was reported in areas CA1 and CA3, we used the Wilcoxon signed-rank test to compare that signal with the corresponding signal in the ipsilateral DG and CA2 to help test for background variation. (The DG and CA2 were selected as there are no reports of ACA inputs to these subareas.)

There were three cases of wild-type mice (C57BL/6J) with 100% of the injection volume within ACA. Of the 44 transgenic mice, 28 had 100% of the tracer within the ACA, making a total of 31 such cases. The other 16 cases had a minimum of 90% of the tracer in the ACA and the remainder in adjacent areas, including the secondary motor area, the prelimbic cortex, and adjacent fibre tracts in various proportions. These 16 cases were, therefore, not considered in the initial analyses. In the WT group all mice were male, whilst in the transgenic group of 28 cases with 100% of the injection in the ACA, 12 were female. In the same transgenic group, 27 of the 28 injections were in the right hemisphere (see *Figure 3—source data 1* for a list of included studies).

## Acknowledgements

This work was supported by Biotechnology and Biological Sciences Research Council grant BB/P001475/1 and a Vandervell Foundation early career fellowship (MTC). SY was a PhD student supported by Wellcome Trust grant 108891/B/15/Z. GMS and ESB were PhD students supported by grant MR/N0137941/1 for the GW4 BIOMED MRC DTP, awarded to the Universities of Bath, Bristol, Cardiff and Exeter from the Medical Research Council (MRC)/UKRI. We gratefully acknowledge the support of the Inger M Lilya and George Simpson Biological Psychiatry Scholarships. Floxed TVA mice were kindly provided by Prof Dieter Saur at Technischen Universität München. We thank Dr E Bubb and Dr A Nelson (University of Cardiff) for access to relevant tracer data. We thank the Spinal Cord Group (University of Glasgow) for access to a confocal microscope.

## Additional information

### Funding

| Funder | Grant reference number | Author |
| --- | --- | --- |
| Biotechnology and Biological Sciences Research Council | BB/P001475/1 | Michael T Craig |
| Medical Research Council | MR/N0137941/1 | Gabriella Margetts-Smith Erica S Brady |
| Wellcome Trust | 108891/B/15/Z | Steliana Yanakieva |

The funders had no role in study design, data collection and interpretation, or the decision to submit the work for publication. For the purpose of Open Access, the authors have applied a CC BY public copyright license to any Author Accepted Manuscript version arising from this submission.

### Author contributions

Lilya Andrianova, Formal analysis, Investigation, Visualization, Methodology, Writing – original draft, Project administration, Writing – review and editing; Steliana Yanakieva, Formal analysis, Funding acquisition, Investigation, Visualization, Methodology, Writing – original draft, Writing – review and editing; Gabriella Margetts-Smith, Erica S Brady, Funding acquisition, Investigation, Writing – review and editing; Shivali Kohli, Investigation, Writing – review and editing; John P Aggleton, Conceptualization, Supervision, Funding acquisition, Writing – original draft, Writing – review and editing; Michael T Craig, Conceptualization, Data curation, Formal analysis, Supervision, Funding acquisition, Investigation, Methodology, Writing – original draft, Project administration, Writing – review and editing

### Author ORCIDs

Gabriella Margetts-Smith http://orcid.org/0000-0002-1885-2661
Michael T Craig http://orcid.org/0000-0001-8481-6709

### Ethics

All UK-based research was carried out in accordance with the UK Animals (Scientific Procedures) Act 1986, and was subject to local ethical review by the Animal Welfare and Ethical Review Board at the University of Exeter or University of Glasgow. All surgical procedures were carried out using aseptic technique under isoflurane anaesthesia, with additional analgesia provided peri- and post-operatively. Every effort was made to minimise animal suffering.

### Decision letter and Author response

Decision letter https://doi.org/10.7554/eLife.77364.sa1
Author response https://doi.org/10.7554/eLife.77364.sa2

## Additional files

### Supplementary files
• Transparent reporting form

### Data availability

All data generated or analysed during this study are included in the manuscript and supporting file. Source data have been provided for Figure 3.

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
