## [Editor Report]

In this important study, the authors attempted to further examine the existence of a potential direct projection from the anterior cingulate cortex to the hippocampus which has important functional implications but is currently supported by only limited evidence. The authors used appropriate and complementary approaches to provide compelling evidence for the conclusion that they found no evidence in support of the existence of this connection.

---

## [Decision Letter]

**Decision letter after peer review:**

Thank you for submitting your article "No evidence from complementary data sources of a direct projection from the mouse anterior cingulate area to the hippocampal formation" for consideration by *eLife*. Your article has been reviewed by 3 peer reviewers, including Mathieu Wolff as Reviewing Editor and Reviewer #1, and the evaluation has been overseen by Laura Colgin as the Senior Editor. The following individual involved in review of your submission has agreed to reveal their identity: Thomas Van Groen (Reviewer #3).

Essential revisions:

1) There is a need to consolidate the retrograde tracing approach. Only a few starter cells were reported with the pseudo-rabies approach while the retrograde AAV used was found to produce off-target labeling, so there is a risk that the entire retrograde approach is not fully compelling. Further analyses (e.g. some quantitative analysis of the number of starter cells) and comments are needed. Relying on more classic retrograde tracers is also an option.

2) Given the nature of the work, all figures must be considerably improved. Higher magnification is needed in many cases and figure captions need to be more comprehensive (see reviewers 1 and 2 for details). Perhaps more illustrations are needed to build stronger cases since that is the only material for the reader to appreciate the current work.

3) The text needs to be more comprehensively expanded. The Results section in particular needs a serious overhaul, to provide much more information on the anatomical data and to compare that with those already reported in the literature (see ref 2 in particular). As a result it is expected that the discussion would also be more thorough, to discuss with greater detail the neuroanatomy and better take into account the limits of the current study (partially depending on how point 1/ is handled). Overall all referee agreed this is an important paper but to have the right impact it needs to be much better presented.

Please refer to individual reviews below for more details.

*Reviewer #1 (Recommendations for the authors):*

I would also suggest to enhance the quality of the figures if possible. These are the only material to consider here and I often found it hard to see the actual labeling, perhaps a higher magnification could help in some cases (Figures 2C/2J) for starter cells; in general, all figures supposed to show labelled cells bodies for retrograde approaches. The meaning of green versus red labeling in figure 2 needs to be explained so that a non-specialist can understand it. Drawing of thalamic nuclei in figure 1D seem a bit arbitrary (Figure 1F seems much more accurate).

*Reviewer #2 (Recommendations for the authors):*

The authors examined the connections of the anterior cingulate cortex (ACA) with the dorsal and ventral hippocampus (CA fields and subiculum) using viral vector techniques in mice and further analyzed data from the Allen Brain Atlas examining these connections. The main intent of the manuscript was to determine the presence or absence of direct ACA-hippocampal projections in light of a previous influential report (Rajasethupath et al., 2015) showing such projections.

1. There has been a growing body of evidence/support for the notion that the hippocampus (HP) projects directly to the medial prefrontal cortex (mPFC), but there are no "direct" return from the mPFC to the HP, and thus mPFC actions on the HP are mediated by intervening structures, mainly by the nucleus reuniens. In possible contrast to this, a recent report by Rajasethupath et al. (2015) described direct projections from the ACA to CA1/CA3 of the HP in mice – with important functional implications. As the authors rightfully point out, if this direct connection was accepted, it could possibly initiate the search for other (misleading) cortical connections to the HP (among other things). Accordingly, the present manuscript serves the very valuable function of disputing a direct ACA input to the hippocampus.

2. A major issue, however, is that this is "all" the manuscript does; that is, it does not provide any additional (or new) information on the projections of the ACA or inputs to the hippocampus – which presumably it could have. In addition, they should have built a stronger case from the existing literature showing a lack of projections from all areas of the mPFC, including ACA, to the hippocampus (see, Heidbreder and Groenewegen, 2003; Vertes, 2004). Although some of this was done in the Discussion, they should expand on this, possibly incorporating it into the Introduction.

3. The figures are generally poor – could be much improved. See below. In the first instance, the labeling (abbreviations) on the figures (and subpanels) is much too small to discern at print size. They need to be enlarged.

Figure 1. Fig, 1A,B: are the graded blue and green scales (lower) meant to show separate injections, this can't be seen on the cortical sections (A and B)? The actual injection shown in Figure 1A (to the right) is quite large and encompasses a significant part of M2 at this level. They should include a sample injection restricted to ACA. There are no micrographs in Figure 1 showing the ventral hippocampus; this should be included. The thalamic labeling with the ACA injections (Figure 1 D,F) is not typical of ACA-thalamic projections. And what are the thalamic nuclei that were labeled with the ACA injections (Figure 1 D,E); this needs to be described in the figure legend and text of the Results. Figure 1H needs to be improved; it is not clear what is being shown -- diagonal band or lateral septal labeling?

Figure 2. It was indicated that the counts of the retrogradely labeled cells were done semi-quantitatively, this needs to be described in greater detail. The started cells (with arrows) of the dorsal and ventral HP need to be shown at higher magnification – even higher than shown in Figure 2J. And very importantly, it seems that the technique only labeled a small percentage of CA1 neurons (judging by the arrows) and if so, this could account relative paucity of retrogradely labeled cells in the structures afferent to the hippocampus, such as the medial septum (Figure 2N). The medial septum should be filled with cells. But this could also account for the lack of labeled cells in ACA (few started cells in CA1), or at least this needs to be addressed. Perhaps the nucleus reuniens should be included as a major afferent to HP. Finally, it is critical to show retrogradely labeled cells at higher magnification, at present all that can be seen are red areas, no individual cells can be distinguished. Lastly, is the PFC in Figure 2P meant to represent the entire prefrontal cortex or the ACA?

Figure 3. When they indicate that the ACA injections from the Allen atlas where 100% or 90% contained within ACA, was this their estimate and if so how was this done. I assume that the density values are from the atlas, but how do they interpret the very low densities (i.e., in CA1, Sub or DG) when they are not zero? Although possibly for present purposes the Allen atlas may be sufficient, it is not comprehensive – as can be judged by the subcortical labeling profiles shown in Figure 3A-C. And the statement that the viral tracing of the Allen atlas is comparable to the transport properties of BDA is not reassuring in that BDA is a poor anterograde tracer.

Figure 4. Very poor, can't see the retrograde cell labeling in cortical or subcortical structures. Like Figure 2 labeled cells need to be shown at high magnification.

4. The description (text) of their anterograde and retrograde tracing experiments in the Results needs to be expanded, only two relatively short paragraphs are devoted to each (anterograde and retrograde experiments) – and much less than the two and half pages devoted to the findings with Allen atlas. For instance, for the anterograde tracing only one sentence was devoted to describing their results (p. 9), essentially only stating that they observed ACA projections to the "retrosplenial cortex and several thalamic nuclei" but no labeling in the hippocampus. The text of the Results should describe the data of Figure 1, panel by panel. The same for the retrograde experiment.

5. The Discussion was rather incomplete. They should provide a more thorough summary of their findings at the outset of the Discussion. While, as they indicate, the Rajasethupath et al. (2015) study is very influential and hence the need to devote considerable attention to it, the Bian et al. (2019) study probably deserves to be discussed in greater detail. For instance, it was only described in the Discussion and not throughout the manuscript; perhaps it should be mentioned in the Introduction. Additionally, it was not well described in the Discussion. The section entitled "Stereotaxic coordinates and the borders of the anterior cingulate cortex" is out of place and not relevant to the Discussion. Some of this could be included in the Introduction or Methods, that is, ACA nomenclature and coordinates. The Discussion should put a greater emphasis on their overall findings – not just the lack of ACA connections with the hippocampus but importantly their verification of other (typical) ACA outputs or HP inputs. Seemingly, this was not done in the Rajasethupath et al. or Bian et al. studies and should have been.

---

## [Author Response]

Essential revisions:1) There is a need to consolidate the retrograde tracing approach. Only a few starter cells were reported with the pseudo-rabies approach while the retrograde AAV used was found to produce off-target labeling, so there is a risk that the entire retrograde approach is not fully compelling. Further analyses (e.g. some quantitative analysis of the number of starter cells) and comments are needed. Relying on more classic retrograde tracers is also an option.

We have carried out quantitative analysis of starter cells in our modified pseudo-rabies approach, which can be seen in Supplementary material, figure 2: supplement 1.

We have also carried out additional retrograde experiment, where we used Fast Blue dye injection into dCA1 to confirm the finding.

2) Given the nature of the work, all figures must be considerably improved. Higher magnification is needed in many cases and figure captions need to be more comprehensive (see reviewers 1 and 2 for details). Perhaps more illustrations are needed to build stronger cases since that is the only material for the reader to appreciate the current work.

We have re-taken all of the representative images, including the supplementary materials, where necessary, using a confocal microscope to drastically improve the quality of the images. We have provided more examples of modified pseudo-rabies tracing in Supplementary material.

3) The text needs to be more comprehensively expanded. The Results section in particular needs a serious overhaul, to provide much more information on the anatomical data and to compare that with those already reported in the literature (see ref 2 in particular). As a result it is expected that the discussion would also be more thorough, to discuss with greater detail the neuroanatomy and better take into account the limits of the current study (partially depending on how point 1/ is handled). Overall all referee agreed this is an important paper but to have the right impact it needs to be much better presented.

We have vastly expanded the descriptive portion of our Results sections to relate our findings to those already known in the literature.

We have also thoroughly expanded the discussion to consider both retrograde and anterograde tracing in wider detail, discuss limitations of our own study and to more thoroughly discuss surrounding literature and potential caveats of viral tracing.

Please refer to individual reviews below for more details.Reviewer #1 (Recommendations for the authors):I would also suggest to enhance the quality of the figures if possible. These are the only material to consider here and I often found it hard to see the actual labeling, perhaps a higher magnification could help in some cases (Figures 2C/2J) for starter cells; in general, all figures supposed to show labelled cells bodies for retrograde approaches. The meaning of green versus red labeling in figure 2 needs to be explained so that a non-specialist can understand it. Drawing of thalamic nuclei in figure 1D seem a bit arbitrary (Figure 1F seems much more accurate).

We have re-made all of the figures now using confocal microscopy, and adding in zoomed in panels to allow the reader to see both the cells in details and the wider tissue slice alike.

We have added a supplementary figure for figure 2 with more injection site examples as well as quantification of starter cell numbers.

We have added a paragraph into the Results section, explaining the meaning of fluorescent signals in this experiment.

We have gone over our annotations of the thalamic nuclei to ensure that all the labels are accurate.

Reviewer #2 (Recommendations for the authors):The authors examined the connections of the anterior cingulate cortex (ACA) with the dorsal and ventral hippocampus (CA fields and subiculum) using viral vector techniques in mice and further analyzed data from the Allen Brain Atlas examining these connections. The main intent of the manuscript was to determine the presence or absence of direct ACA-hippocampal projections in light of a previous influential report (Rajasethupath et al., 2015) showing such projections.Comments (in no particular order).3. The figures are generally poor – could be much improved. See below. In the first instance, the labeling (abbreviations) on the figures (and subpanels) is much too small to discern at print size. They need to be enlarged.Figure 1. Fig, 1A,B: are the graded blue and green scales (lower) meant to show separate injections, this can't be seen on the cortical sections (A and B)? The actual injection shown in Figure 1A (to the right) is quite large and encompasses a significant part of M2 at this level. They should include a sample injection restricted to ACA. There are no micrographs in Figure 1 showing the ventral hippocampus; this should be included. The thalamic labeling with the ACA injections (Figure 1 D,F) is not typical of ACA-thalamic projections. And what are the thalamic nuclei that were labeled with the ACA injections (Figure 1 D,E); this needs to be described in the figure legend and text of the Results. Figure 1H needs to be improved; it is not clear what is being shown -- diagonal band or lateral septal labeling?

Figure 1A and 1B show cartoon overlays of blue and green injection spreads, with increasing ‘strength’ of colour indicating the number of animals that had fluorophore expression in a particular region. This allows us to present a summary of injection sites across a larger number of animals than can be shown. We have described this more explicitly in the text to aid the reader. Whereas the representative examples show only one injection spread, as they are images of brain tissue.

Figure 1A shows injection sites that spread into M2 as these correspond to the injections made by the paper that we were trying to replicate, thus it was essential to include in the manuscript. Figure 1B, by contrast, presents more optimised injections that were restricted to prefrontal areas. The most important point is that, even when virus spreads beyond ACA to regions such as M2, we still see no efferent axons in hippocampus (HPC).

We have made sure that all of the abbreviations are now included in the figure legend.

We have not included images of ventral hippocampus (vHPC) as no claims were made about ACA to vHPC in the original paper that we tried to replicate and, while our rabies tracing shows no input from vHPC from ACA, we have omitted showing the vHPC images in this figure to aid clarity and prevent the figure becoming too crowded.

Figure 2. It was indicated that the counts of the retrogradely labeled cells were done semi-quantitatively, this needs to be described in greater detail. The started cells (with arrows) of the dorsal and ventral HP need to be shown at higher magnification – even higher than shown in Figure 2J. And very importantly, it seems that the technique only labeled a small percentage of CA1 neurons (judging by the arrows) and if so, this could account relative paucity of retrogradely labeled cells in the structures afferent to the hippocampus, such as the medial septum (Figure 2N). The medial septum should be filled with cells. But this could also account for the lack of labeled cells in ACA (few started cells in CA1), or at least this needs to be addressed. Perhaps the nucleus reuniens should be included as a major afferent to HP. Finally, it is critical to show retrogradely labeled cells at higher magnification, at present all that can be seen are red areas, no individual cells can be distinguished. Lastly, is the PFC in Figure 2P meant to represent the entire prefrontal cortex or the ACA?

We have added a supplementary figure with additional examples of injection site, made with a confocal microscope as well as quantification of the number of starter cells per image as well as estimate of the total number of starter cells present in the brain parenchyma. The starter cells are now easier to see with the updated images. We have included more higher magnification images of the retrogradely-labelled neurons now. The starter cells density could be considered a little on the low side, but we have added quantitative data, classical retrograde tracing, and more discussion of starter cell density as a potential caveat.

NRe is not included in the images as we believe that it only targets inhibitory interneurons in hippocampus proper, but does target pyramidal cells in prosubiculum and subiculum (as well as EC and PFC). Please refer to our preprint for more information – doi.org/10.1101/2021.09.30.462517. This finding was originally contentious (and the submission was rejected by *eLife*) but as can be seen from the studies that cite our preprint, evidence is growing in the literature that NRe projections to CA1 primarily or exclusively targets interneurons.

In figure 2P PFC includes the following sub-regions of the prefrontal cortices, including ACC, PL, IL.

Figure 3. When they indicate that the ACA injections from the Allen atlas where 100% or 90% contained within ACA, was this their estimate and if so how was this done. I assume that the density values are from the atlas, but how do they interpret the very low densities (i.e., in CA1, Sub or DG) when they are not zero? Although possibly for present purposes the Allen atlas may be sufficient, it is not comprehensive – as can be judged by the subcortical labeling profiles shown in Figure 3A-C. And the statement that the viral tracing of the Allen atlas is comparable to the transport properties of BDA is not reassuring in that BDA is a poor anterograde tracer.

In figure 3 the 90-100% of the virus refers to the proportion of volume of virus that is restricted to the ACC area specifically. This is the Allen Atlas estimate, not ours, based on optical analyses. The very low counts reflect the almost inevitable background that will be present. This is the reason why we used counts from areas known NOT to receive ACA inputs to give a background control level. These same background measures helped to verify the lack of any transport to the target hippocampal regions.

We have modified the statement regarding the viral tracing method and BDA, reflecting the exact statements in the Allen atlas. Those statements confirm that AAV was at least as efficient as, and more specific than, the classic anterograde tracer biotinylated dextran amine. In other words, the viral tracer is superior to BDA, while the comparison helped to confirm that it does not create false positive.

Figure 4. Very poor, can't see the retrograde cell labeling in cortical or subcortical structures. Like Figure 2 labeled cells need to be shown at high magnification.

We have taken all of the representative images using a confocal microscope, thus drastically improving the quality of the images, so individual cells can now be seen.

4. The description (text) of their anterograde and retrograde tracing experiments in the Results needs to be expanded, only two relatively short paragraphs are devoted to each (anterograde and retrograde experiments) – and much less than the two and half pages devoted to the findings with Allen atlas. For instance, for the anterograde tracing only one sentence was devoted to describing their results (p. 9), essentially only stating that they observed ACA projections to the "retrosplenial cortex and several thalamic nuclei" but no labeling in the hippocampus. The text of the Results should describe the data of Figure 1, panel by panel. The same for the retrograde experiment.

We have added a thorough description of the data presented on figure 1 and figure 4 with the retrograde labelling.

5. The Discussion was rather incomplete. They should provide a more thorough summary of their findings at the outset of the Discussion. While, as they indicate, the Rajasethupath et al. (2015) study is very influential and hence the need to devote considerable attention to it, the Bian et al. (2019) study probably deserves to be discussed in greater detail. For instance, it was only described in the Discussion and not throughout the manuscript; perhaps it should be mentioned in the Introduction. Additionally, it was not well described in the Discussion. The section entitled "Stereotaxic coordinates and the borders of the anterior cingulate cortex" is out of place and not relevant to the Discussion. Some of this could be included in the Introduction or Methods, that is, ACA nomenclature and coordinates. The Discussion should put a greater emphasis on their overall findings – not just the lack of ACA connections with the hippocampus but importantly their verification of other (typical) ACA outputs or HP inputs. Seemingly, this was not done in the Rajasethupath et al. or Bian et al. studies and should have been.

We have now substantially expanded and re-written the Discussion section to discuss the positive controls (known projections that we found from ACA), considered Bian in greater detail, and to more fully discuss different studies looking at anterograde and retrograde tracing from ACA and HPC, respectively. We have also incorporated some whole brain single cell PFC projectome data from a recent publication (Gao et al., 2022, https://www.nature.com/articles/s41593-022-01041-5) which further supported our findings.

The discussion of ACA borders has been edited and moved to the methods section.